# Sirolimus Prolongs Survival after Living Donor Liver Transplantation for Hepatocellular Carcinoma Beyond Milan Criteria: A Prospective, Randomised, Open-Label, Multicentre Phase 2 Trial

**DOI:** 10.3390/jcm9103264

**Published:** 2020-10-12

**Authors:** Kwang-Woong Lee, Seong Hoon Kim, Kyung Chul Yoon, Jeong-Moo Lee, Jae-Hyung Cho, Suk Kyun Hong, Nam-Joon Yi, Sung-Sik Han, Sang-Jae Park, Kyung-Suk Suh

**Affiliations:** 1Department of Surgery, Seoul National University Hospital, 101 Daehak-no, Jongno-gu, Seoul 110-744, Korea; freefine@daum.net (K.C.Y.); lulu5050@naver.com (J.-M.L.); bunny174@naver.com (J.-H.C.); nobel1210@naver.com (S.K.H.); gsleenj@hanmail.net (N.-J.Y.); kssuh2000@gmail.com (K.-S.S.); 2Center for Liver Cancer, National Cancer Center, 323 Ilsan-ro, Ilsandong-gu, Goyang-si 410-769, Gyeonggi-do, Korea; kshlj@hanmail.net (S.H.K.); sshan@ncc.re.kr (S.-S.H.); spark@ncc.re.kr (S.-J.P.)

**Keywords:** sirolimus, hepatocellular carcinoma, transplantation, survival

## Abstract

Sirolimus (SRL) has been reported to benefit patients undergoing liver transplantation (LT) for hepatocellular carcinoma (HCC). This study aimed to compare SRL with tacrolimus (TAC) in living-donor LT (LDLT) recipients beyond the Milan criteria. This study was initially designed to enrol 45 recipients who underwent LDLT for HCC beyond the Milan criteria. At 1 month after LT, the patients were randomly assigned to either SRL or TAC-based treatment, with both groups receiving mycophenolate mofetil. The primary outcome was three-year recurrence-free survival (RFS) and the secondary outcome was overall survival (OS). A total of 42 patients completed the study. HCC recurrence occurred in 8 of 22 (36.4%) patients in the SRL group and in 5 of 22 (25%) patients in the TAC group. No differences in RFS and OS were found between the two groups in simple comparison. The type of immunosuppressant remained a nonsignificant factor for recurrence in multivariate analysis; however, SRL significantly prolonged OS (TAC hazard ratio: 15 [1.3–172.85], *p* = 0.03) after adjusting for alpha-fetoprotein and positron emission tomography standardised uptake value ratio (tumour/background liver). In conclusion, SRL does not decrease HCC recurrence but prolongs OS after LDLT for HCC beyond the Milan criteria.

## 1. Introduction

Liver transplantation (LT) is an outstanding treatment option for patients with hepatocellular carcinoma (HCC). Complete resection of the tumour and underlying oncogenic cirrhotic liver provides superior oncologic outcomes. Several morphological selection criteria for LT, including the Milan criteria and the University of California San Francisco (UCSF) criteria, have been proposed because patients with tumour recurrence after LT show a poor prognosis. However, some patients beyond these criteria have shown good outcomes, thereby expanding the indications of LT, especially in living-donor LT (LDLT) [1]. Biological criteria including alpha-fetoprotein (AFP) and protein induced by vitamin K absence/antagonist (PIVKA)-II levels, positron emission tomography (PET) positivity, and treatment response to locoregional therapy can also be used to select optimal LT candidates outside the conventional morphological criteria [2,3]. Nonetheless, expansion of the conventional selection criteria for LT recipients inevitably leads to an increased rate of tumour recurrence. Therefore, treatments to reduce the risk of recurrence, while prolonging survival rates, are needed.

The impact of immunosuppression on HCC recurrence after LT has been reported in the literature. The higher the exposure to calcineurin inhibitor (CNI), the higher the risk of post-LT HCC recurrence [4]. Mammalian target of rapamycin (mTOR) inhibitors inhibit HCC growth in vitro and in vivo animal experiments [5]. Therefore, mTOR inhibitors may have a favorable effect in reducing the incidence of post-LT HCC recurrence. According to several retrospective studies [6], meta-analyses [7] and post-hoc assessments of randomised trials [8], mammalian targeting of rapamycin (mTOR) inhibitors such as sirolimus (SRL) have been considered to be promising immunosuppressants for reducing the recurrence of HCC after LT [9]. However, whether this benefit is due to the direct effects of mTOR inhibitors remains unclear, as previous studies compared an mTOR inhibitor plus a low dose of tacrolimus (TAC) with a high dose of TAC. Furthermore, a recent international prospective multicentre study (SRL in Liver Transplant Recipients with HCC study [SILVER]) that compared an mTOR inhibitor-based therapy with an mTOR inhibitor-free therapy reported a negative result [10]. Therefore, the role of SRL in advanced HCC remains ambiguous. Well-designed direct comparative studies, especially in patients who have undergone LDLT, are needed to assess the effects of SRL in these patients. Therefore, this study aimed to directly compare the outcomes of SRL and TAC in LDLT recipients beyond the Milan criteria.

## 2. Materials and Methods

### 2.1. Study Design

This prospective, randomised, multicentre phase 2 trial compared the oncological outcomes of SRL versus TAC in patients undergoing LDLT beyond the Milan criteria from two tertiary referral centres in the Republic of Korea (Seoul National University Hospital and National Cancer Center). This study was approved by the Seoul National University Hospital Institutional Review Board (approval no. H-1004-052-316) and National Cancer Center Institutional Review Board (approval no. NCCCTS-08-368) and is registered at ClinicalTrials.gov (NCT01374750). All methods employed in this study were performed in accordance with the relevant guidelines and regulations after obtaining informed consent.

Patients were included if they satisfied the following criteria: (i) age ≥ 18 years with written consent, (ii) body weight ≥40 kg, and (iii) histologically proven HCC beyond the Milan criteria. Tumour size included any necrotic portion if a viable portion remained. Nodules with complete necrosis were not considered tumour nodules. Multiorgan transplant and DDLT recipients, as well as patients with tumour thrombosis in major vessels (Vp1 or Vp2) or extrahepatic metastasis were excluded.

### 2.2. Sample Size Calculation

The sample size was calculated using the expected 3-year RFS assumed from previous reports on the 5-year OS of patients with HCC beyond the Milan criteria because the 3-year RFS was similar to the 5-year OS. According to an ‘HCC forecast chart’, the 5-year OS was inevitably wide ranging, from 15% to 70%, because there was no upper limit [11]. The 5-year OS of patients with HCC beyond the Milan criteria but within the UCSF criteria has been reported to range from 38% to 93% [12]. Beyond the UCSF criteria, up to major vessel invasion (segmental branch) will be included; therefore, the survival rate will be lower. We assumed that the 3-year RFS of the TAC group will be 30% and that of the SRL group will be 60%. The RFS distributions of the two groups were compared using a two-sided log-rank test with a 0.05 significance level and 0.8 statistical power. Using the calculation offered by Schulz and Grimes [13], the estimated number of patients in each group will be 20. With a loss to follow-up rate of about 13%, a total of 45 patients was deemed to be required.

### 2.3. Treatment Protocol

Immunosuppression before randomisation consisted of a quadruple regimen that included basiliximab, TAC, mycophenolate mofetil (MMF), and steroid, with the latter tapered off within 1 month of LT. At the time of randomisation, all patients were being treated with TAC and MMF. One month after LT, the patients were randomised into two groups: receiving either TAC or SRL. To balance the recurrence risk, patients were stratified by microvascular invasion and serum AFP level >400 ng/mL. Patients in the TAC group were maintained on TAC and MMF, whereas patients in the SRL group were switched to SRL and MMF. The target levels of TAC and SRL were similar, ranging from 6 to 12 ng/mL. The level of MMF was not monitored but the dosage was 0.5–1.5 g/day. 

After recurrence in the TAC group, incorporation of SRL was allowed according to the clinician’s decision.

### 2.4. Follow-up and Documentation

In the first year after LDLT, all patients were followed up at 1-month intervals of up to 6 months, then at a 2-month interval up to the first year. Thereafter, the patients were followed up every 3 months. Imaging workup for surveillance of HCC recurrence, including liver computed tomography, chest computed tomography, and bone scan, was performed at 3-month intervals in the first year and at 6-month intervals thereafter. Doppler ultrasonography was performed before discharge in both groups, and at 2 months after LT in the SRL group.

Data on serious complications or adverse events (such as graft failure, biopsy-proven acute cellular rejection, serious infection, hepatic artery occlusion, HCC recurrence, and death) and minor complications or adverse events (such as post-transplant diabetes, dyslipidaemia, renal impairment, hypertension, minor infectious complications, gastrointestinal adverse effects, and hematologic adverse effects) were recorded on electronic case report forms at regular visits.

### 2.5. Study Endpoints

The primary outcome of this study was the 3-year RFS and the secondary outcome was the 3-year OS and the safety of SRL.

### 2.6. Statistical Analysis

Nonparametric continuous variables are presented as median with range, and parametric continuous variables are presented as mean ± standard deviation. Categorical variables are described as the number and proportion. Mann-Whitney U test and Student’s *t* test were used for nonparametric and parametric variables, respectively. The χ^2^ test was used for categorical variables. The DFS and OS rates were estimated with the Kaplan-Meier method and compared using a log-rank test. Cox regression using a forward likelihood-ratio selection model was used for multivariate analysis of the risk factors for RFS and OS. The cut-off values of AFP and PET SUV ratio were decided using receiver operating characteristic curve analysis. A probability (*p*) value of < 0.05 was considered significant.

## 3. Results

### 3.1. Patient Enrolment

A total of 45 patients were enrolled from 2010 to 2016. Three patients withdrew consent during the follow-up period. Finally, 42 patients were enrolled. Of these patients, 20 were randomly assigned to the SRL group and 22 to the TAC group. All patients completed the 3-year follow-up (Figure 1).

### 3.2. Clinicopathological Findings

The clinicopathological findings of the two groups are shown in Table 1. Patients in the TAC group were slightly older, although there was no significant difference in most preoperative characteristics. The cold ischemic time was 15 min longer in the SRL group; however, most intraoperative and pathological factors were comparable. The rates of microvascular invasion in the TAC and SRL groups were 23% and 25%, respectively. The details of immunosuppression were presented in Table 1. The most common dosage of MMF was 1000 mg/day in both groups.

### 3.3. Recurrence-Free Survival (RFS) and Overall Survival (OS)

Five (22.7%) patients in the TAC group and eight patients in the SRL group experienced tumour recurrence within three years. The three-year RFS rates in the TAC and SRL groups were 77.3% and 60%, respectively, whereas the three-year OS rates were 81.8% and 77%, respectively. No significant intergroup differences in the three-year RFS and OS rates were found (Figure 2). The tumour-bearing OS (from tumour recurrence to death) of the SRL group was significantly longer than that of the TAC group among 13 patients with tumour recurrence after LT (Figure 3).

### 3.4. Risk Factors for Survival

Multivariate analysis showed that serum AFP level >150 ng/mL and PET standardised uptake value (SUV) ratio (tumour/background liver) >1.15 were significant risk factors for both RFS and OS. Treatment with SRL significantly prolonged OS (hazard ratio 15.0, 95% confidence interval 1.302–172.8, *p* = 0.03) but did not affect RFS (Table 2 and Table 3).

Recipient age, recipient sex, Child-Pugh-Turcotte score, model for end-stage liver disease score, donor sex, donor age, cold ischemic time, warm ischemic time, maximum pathologic size 5 cm, Edmonson and Steiner grade 1–2 vs. 3–4, graft-to-recipient weight ratio 0.8, and treatment group were not significant.

Recipient age, recipient sex, Child-Pugh-Turcotte score, model for end-stage liver disease score, donor sex, donor age, cold ischemic time, warm ischemic time, maximum pathologic size 5 cm, Edmonson and Steiner grade 1–2 vs. 3–4, and graft-to-recipient weight ratio 0.8 were not significant.

### 3.5. Adverse Events and Complications

The rates of wound complication and dyslipidaemia tended to be higher in the SRL group; however, the difference was not statistically significant. One patient in the SRL group experienced acute cellular rejection, and no patient experienced hepatic artery thrombosis in both groups (Table 4).

### 3.6. Changes in Estimated Glomerular Filtration Rate (eGFR)

The eGFR was calculated using the Modification of Diet in Renal Disease equation [6]. A trend of a renal-sparing effect was observed in the SRL group compared with the TAC group, but the difference did not reach statistical significance (*p* = 0.08) (Table 4).

## 4. Discussion

HCC accounts for around 60% of adult LDLTs, including a higher proportion of patients who undergo LDLT beyond the Milan criteria [1]. At present, HCC beyond the Milan criteria accounts for 30–40% of LDLT for HCC cases at Seoul National University Hospital. The expansion of indications for LDLT inevitably results in a higher tumour recurrence rate [14]. In our series, about 31% of patients experienced tumour recurrence. Therefore, effective measures are needed to reduce recurrence and prolong survival.

mTOR inhibitors have shown beneficial effects in vitro and in vivo [5]. As the mTOR pathway is upregulated in tumour cells, mTOR inhibitors can reduce tumour growth by inhibiting angiogenesis and inducing the apoptosis of tumour cells. Therefore, mTOR inhibitors can potentially promote the prevention and reduction of HCC recurrence in patients after LT. Several clinical studies have also reported that mTOR inhibitors have beneficial effects. A recent meta-analysis reported that the use of these agents can prevent or treat HCC recurrences in patients who underwent LT for HCC [8]. Most of the studies included in that meta-analysis, however, had a retrospective design and enrolled patients treated with combinations of TAC and an mTOR inhibitor. A post-hoc analysis of the prospective randomised H2304 trial study showed that RFS was better in patients treated with everolimus plus a lower dose of TAC than in those treated with TAC alone, although the difference was not statistically significant [8]. As these studies compared an mTOR inhibitor plus a low dose of TAC with a high dose of TAC, it is unclear whether the benefits for patients were due to a direct effect of the mTOR inhibitor or an indirect effect of reduced TAC dose, indicating the need for direct comparisons between mTOR inhibitors and TAC.

The SILVER trial showed that SRL can improve the three to five-year RFS and OS rates in low-risk patients within the Milan criteria [10]. However, SRL did not improve the long-term RFS > five years after LT, although it tended to benefit low-risk patients. The major limitations of that study were the categorisation of risk with only the Milan criteria and the inclusion of patients who underwent both deceased-donor LT (DDLT) and LDLT, which can lead to a different outcome. Furthermore, the immunosuppressive regimens in the SILVER trial considerably varied because many centres participated.

A recent study from China reported that SRL improved OS [15]. In that trial, 142 patients with HCC who underwent LT were treated with (*n* = 62) or without (*n* = 80) SRL. The RFS rates did not significantly differ between the two groups. However, the median tumour-bearing survival after HCC recurrence, which correlates with OS, was significantly longer in patients treated with SRL (12 months, range 3–24 months) than in the control group (8 months, range 6–22 months), indicating that SRL can prolong survival after HCC recurrence. However, that study was a retrospective analysis and did not use biological criteria such as the AFP level.

To our knowledge, this study is the first prospective direct comparison in patients who underwent LDLT for HCC under a uniform immunosuppressive regimen. While the RFS and OS between the SRL and TAC groups were not different in simple comparison analysis, it may be due to both the small number of patients and the lack of consideration of important risk factors such as PET positivity at the time of stratification or survival analysis. Multivariate analysis including potential risk factors (Table 2 and Table 3) and tumour-bearing survival analysis after recurrence (Figure 3) were performed to overcome these limitations. Despite these additional analyses, no benefit of early initiation of SRL in reducing recurrence was found; however, SRL clearly showed a benefit in prolonging the OS. In particular, SRL significantly prolonged survival after recurrence compared with TAC.

The cellular mechanism by which SRL prolongs survival without reducing tumour recurrence is unclear. Rapamycin inhibits cell signalling through the phosphatidylinositol 3-kinase/Akt/mTOR pathway, which is involved in cell growth, apoptosis, metabolism, and differentiation [16]. Our previous in vitro and animal experiments showed that mTOR inhibitors inhibit cell growth without reducing the cell number [5]. TAC has been found to induce the expression of transforming growth factor-β, which is important for epithelial-mesenchymal transition [17]. mTOR inhibitors have been found to reduce epithelial-mesenchymal transition, potentially preventing cells from metastatic progression or aggressive transformation [18]. Therefore, in this study, SRL may not have a cytotoxic effect on circulating cancer cells at the time of LT, but may attenuate the aggressiveness of the cancer cell.

Early initiation of SRL is important for prolonging survival in cases with recurrence. The patients in our study who developed HCC recurrence, even those in the TAC group, were allowed to start SRL. All five recurred cases in the TAC group were switched to SRL. However, tumour-bearing survival was better in patients initially randomised to SRL than in those started on TAC and switched to SRL after tumour recurrence (Figure 3).

Despite doubts about the safety of early SRL for immunosuppression after LT, we observed no significant differences in major complications between the TAC and SRL groups. While the frequency of wound complications and dyslipidaemia tended to be higher in the SRL group, the differences were not significant. Furthermore, only one patient experienced acute cellular rejection (ACR) and none experienced hepatic artery thrombosis. However, there is concern of ACR in CNI free regimen. In particular, from the painful experience of higher ACR in the early CNI withdrawal group in H2304 study [19], mTOR inhibitors ± MMF without CNI is still considered to be a risky regimen. However, we experienced satisfactory outcomes in the SRL+MMF regimen. The incidence of SRL-associated complications, including ACR, was very low, with the most common adverse effects being manageable or resolved after drug reduction. Furthermore, the SRL+MMF regimen was the best immunosuppressive regimen in terms of the inhibition of HCC growth in our animal experiment [5].

Considering the tolerable adverse effects in this study, mTOR inhibitor-based immunosuppression can be considered in most HCC recipients. However, mTOR inhibitor-based immunosuppression seems only beneficial for recurred cases in our study. Furthermore, one meta-analysis reported that conversion to an mTOR inhibitor was associated with a higher risk of acute rejection (relative risk [RR], 1.76) and study discontinuation due to adverse events (RR, 2.17) up to 1 year after the conversion from calcineurin inhibitor to mTOR inhibitor-based immunosuppression [20]. Therefore, by balancing the benefit and potential risk of adverse events, an mTOR inhibitor regimen can be selectively recommended to patients at a high risk of tumour recurrence. Preoperatively predicting the exact recurrence rate will be the most important issue. While all patients in our study were outside the Milan criteria, indicating that they were at a high risk, the overall recurrence rate was about 30%, which was lower than expected. The Milan criteria have some limitations, as they are solely based on morphological characteristics. Therefore, the definition of high-risk patients who may benefit from treatment with an mTOR inhibitor after LT should be refined across the Milan criteria, using combinations of preoperative biological and pathological markers (AFP, PIVKA, PET positivity, and microvascular invasion) [2,3,21].

LDLT can be associated with a higher incidence of HCC recurrence after LT than DDLT [21]. Therefore, the type of graft can interfere with the outcome. Different from previous studies that included both LDLT and DDLT cohorts, this study only focused on LDLT and may be less affected by that kind of bias. Nevertheless, the question of whether the conclusion of our study can be expanded to the outcomes of DDLT remains unresolved. Theoretically, it can be expanded to DDLT because, although the recurrence rate itself can be affected by the type of graft, the benefit of mTOR inhibitor treatment was only confirmed in recurred cases and therefore was not associated with the recurrence rate. However, further studies are needed to accurately answer this question.

This study had several limitations. The study population was small (only 42 patients). To compensate for this limitation, multivariate analyses adjusting for significant risk factors were performed. We believe that this phase 2 study including a small cohort provides the clue or tendency with respect to the role of mTOR inhibitor treatment after LT for HCC. Future multicentre studies with larger study populations of high-risk patients defined by biological markers are needed to confirm our findings.

## Figures and Tables

**Figure 1 jcm-09-03264-f001:**
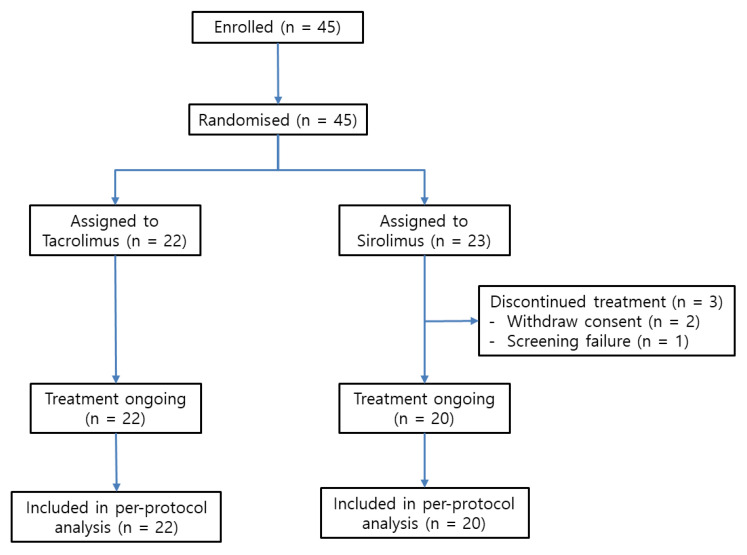
Study flow (CONSORT diagram).

**Figure 2 jcm-09-03264-f002:**
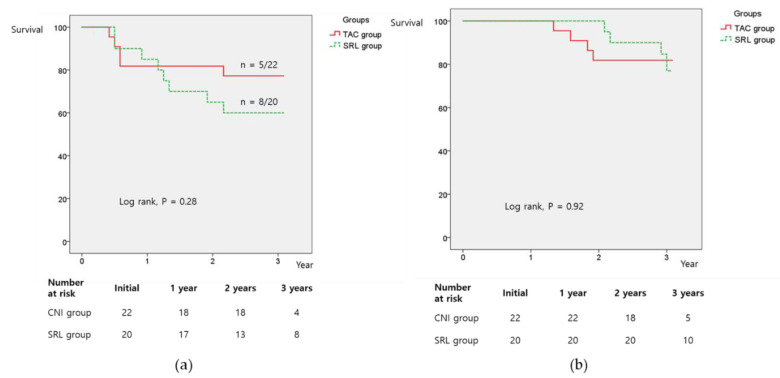
Kaplan-Meier analysis of (**a**) recurrence-free survival and (**b**) overall survival of patients in the sirolimus (SRL) and tacrolimus (TAC) groups. No significant differences were found between the two groups.

**Figure 3 jcm-09-03264-f003:**
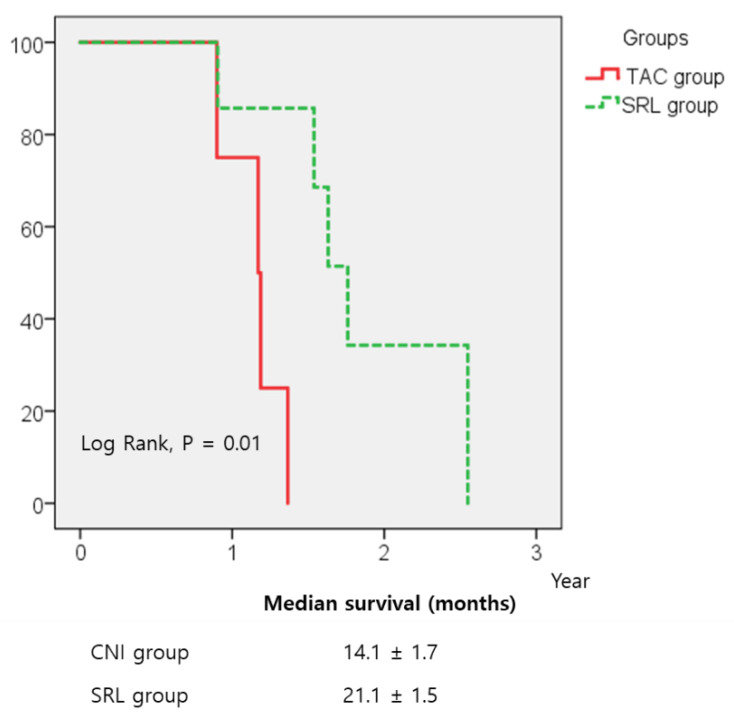
Tumour-bearing survival after recurrence in 13 recurred cases. Early initiation of sirolimus (SRL) at 1 month after living-donor liver transplantation significantly prolonged survival after recurrence compared with tacrolimus (TAC) (*p* = 0.01).

**Table 1 jcm-09-03264-t001:** Clinicopathologic findings in the TAC and SRL groups.

Variable	Categorical Value	TAC (*n* = 22)	SRL (*n* = 20)	*p* Value
**Preoperative factors**				
Recipient Sex	Male	21 (95%)	19 (95%)	1.00
Age (Years)	Mean ± SD	56.86 ± 6.40	51.95 ± 7.63	0.03
Hypertension	*n* (%)	5 (22.7)	4 (20.0)	1.00
DM	*n* (%)	6 (27.3)	8 (40.0)	0.52
BMI (kg/m^2^)	Mean ± SD	24.23 ± 3.05	25.04 ± 3.79	0.45
MELD Score	Mean ± SD	11.18 ± 9.10	8.95 ± 5.35	0.35
CPT Score, *n* (%)	*n* (%)			0.01
A		13 (59.1)	10 (50.0)	
B		3 (13.6)	10 (50.0)	
C		6 (27.3)	0 (0)	
Serum Creatinine (mg/dL)	Median	0.93 (0.58–1.8)	0.86 (0.57–1.3)	0.27
Donor Sex	Male	13 (59.1%)	14 (70%)	0.53
Donor Age	Median	32 (18–53)	25.5 (17–53)	0.36
Primary Liver Disease	*n* (%)			0.40
HBV		17 (77.3)	18 (90.0)	
HCV		1 (4.5)	1 (4.2)	
Alcoholism		3 (13.6)	0 (0)	
Others		1 (13.6)	1 (4.2%)	
Pre-LT Treatment	*n* (%)			
TACE		16 (72.7)	15 (62.5)	0.46
RFA		2 (9.1)	3 (15.0)	0.66
PEIT		2 (9.1)	1 (8.3)	1.00
Surgery		3 (13.6)	3 (12.5)	1.00
AFP (ng/mL)	Mean ± SD	317.93 ± 1274.23	706 ± 2916.97	0.57
	Median	18.55 (2.7–6010)	15.3 (2.4–14,070)	0.49
≤150	*n* (%)	19 (86.4)	16 (80.0)	0.69
>150	*n* (%)	3 (13.6)	4 (20.0)	
≤400	*n* (%)	21 (95.5)	17 (85%)	0.27
>400	*n* (%)	1 (4.5)	3(15%)	
PET SUV ratio	Median	1.02 (0.91–1.68)	1.17 (0.82–3.39)	0.35
≤1.15		13 (76.5)	8 (50)	0.16
>1.15		4 (23.5)	8 (50)	
**Intraoperative factors**				
GRWR	Mean ± SD	1.18 ± 0.31	1.14 ± 0.28	0.69
Cold Ischemia Time (min)	Mean ± SD	63.3 ± 26.72	78.98 ± 21.29	0.05
Warm Ischemia Time (min)	Mean ± SD	33.33 ± 9.71	29.99 ± 8.02	0.23
EBL (Ml)	Median	1475 (400–23,000)	1625 (500–12,000)	0.68
RBC Transfusion (Units)	Median	0.5 (0–36)	2 (0–24)	0.61
Pathologic Factors				
Tumour Number	Median	3.5 (1–8)	4.5 (1–11)	0.33
Tumour Maximum Size (cm)	Median	3.8 (0.9–8)	3.65 (2–10)	0.55
Tumour Sum Size (cm)	Mean ± SD	7.59 ± 4.68	9.01 ± 3.44	0.25
Microvascular Invasion	*n* (%)	5 (22.7)	5 (25.0)	1.00
Portal Vein Invasion	*n* (%)	0 (0)	0 (0)	NA
Bile Duct Involvement	*n* (%)	1 (4.5)	0 (0)	1.00
Mean tumour necrosis after TACE	Median	61.6 (0–100)	31 (0–99)	0.26
ES grade, *n* (%)				0.13
1		1 (4.5)	0 (0)	
2		4 (19.2)	9 (45.0)	
3		12 (54.5)	10 (50.0)	
4		5 (22.7)	1 (5.0)	
**Immunosuppression** Trough Level of TAC or SRL	Median			
3 Months		5.1 (0.6–12.6)	4.5 (1.6–15.1)	
6 Months		8.5 (4.9–15.2)	6.8 (2–30)	
MMF dosage at 3 months	*n* (%)			
~500 mg/day		0	1 (5.0)	
1000 mg/day		17 (77.3)	15 (75.0)	
1500 mg/day		5 (22.7)	4 (20.0)	
**HCC related outcome**				
Time to LT	Median	17 (0–107)	2 (0–108)	0.08
Event of Recurrence	*n* (%)	8 (36.4)	5 (25)	0.43
Time to Recurrence	Median	7 (5–31)	15 (6–45)	0.46
Event of Death	*n* (%)	7 (31.8)	2 (10)	0.14

TAC, tacrolimus; SD, standard deviation; SRL, sirolimus; DM, diabetes mellitus; BMI, body mass index; MELD, model for end-stage liver disease; CPT, Child-Pugh-Turcotte; HBV, hepatitis B; HCV, hepatitis C; LT, liver transplantation; TACE, transarterial chemoembolisation; RFA, radiofrequency ablation; PEIT, percutaneous ethanol injection therapy; AFP, alpha-fetoprotein; PET, positron emission tomography; SUV, standardised uptake value; GRWR, graft-to-recipient weight ratio; EBL, estimated blood loss; RBC, red blood cell; ES, Edmondson and Steiner; NA, not available.

**Table 2 jcm-09-03264-t002:** Multivariate analysis of risk factors for recurrence-free survival.

Variable		HR (95% CI)	*p* Value
AFP (ng/mL)	≤150		
	>150	4.21 (1.26–14.08)	0.02
PET Positivity (Tumour/Background SUV Ratio)	≤1.15		
	>1.15	7.13 (2.18–24.55)	0.01

HR, hazard ratio; CI, confidence interval; AFP, alpha-fetoprotein; PET, positron emission tomography; SUV, standardised uptake value.

**Table 3 jcm-09-03264-t003:** Multivariate analysis of risk factors for overall survival.

Variable		HR (95% CI)	*p* Value
AFP (ng/mL)	≤150		
	>150	35.23 (3.29–377.63)	<0.01
PET Positivity (Tumour/Background SUV Ratio)	≤1.15		
	>1.15	28.03 (2.67–293.95)	<0.01
Treatment Group	Sirolimus		
	Tacrolimus	15.00 (1.30–172.85)	0.03

HR, hazard ratio; CI, confidence interval; AFP, alpha-fetoprotein; PET, positron emission tomography; SUV, standardised uptake value.

**Table 4 jcm-09-03264-t004:** Number and distribution of relevant adverse events and eGFR.

Adverse Event.		TAC (*n* = 22)	SRL (*n* = 20)	
General	Weight Loss	2 (9.1%)	0	0.489
	Oedema of Both Legs	0	1 (4.2%)	0.476
	Wound Complication	0	2 (10%)	0.221
	Wound Lymphocele	0	1 (4.2%)	0.476
	Umbilical Hernia	0	1 (4.2%)	0.476
Cardiovascular	Haemolytic Anaemia	1 (4.5%)	1 (4.2%)	1.000
	HTN	2 (9.1%)	2 (10%)	1.000
	Dyslipidaemia	0	3 (15%)	0.099
Gastrointestinal	LFT Abnormality	1 (4.5%)	0	1.000
	Diffuse Fatty Liver	1 (4.5%)	0	1.000
	Biliary Stricture	2 (9.1%)	2 (10%)	1.000
	Oral Mucositis	0	2 (8.3%)	0.221
	Diarrhoea	1 (4.5%)	1 (4.2%)	1.000
Dermatologic	Scaled Skin, Rash	1 (4.5%)	2 (8.3%)	0.598
	Hyperpigmentation	1 (4.5%)	0	1.000
Infection	Herpes Zoster	0	1 (5%)	0.476
	Pulmonary Tb	0	1 (5%)	0.476
Radiation- and Nexavar-Related Symptom (Diarrhoea, Hand-Foot Syndrome)		0	1 (5%)	0.476
eGFR (MDRD)				
	Initial	92.3 ± 27.7	103.1 ± 30.1	0.24
	1 Year	80.9 ± 28.9	95.8 ± 23.2	0.08
	2 Years	80.5 ± 27.5	95.6 ± 26.7	0.09
	3 Years	74.9 ± 25.0	90.2 ± 25.3	0.08

eGFR, estimated glomerular filtration rate; MDRD, Modification of Diet in Renal Disease equation; TAC, tacrolimus; SRL, sirolimus; HTN, hypertension; LFT, liver function test; Tb, tuberculosis.

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
