# Peer review of "Sirolimus Prolongs Survival after Living Donor Liver Transplantation for Hepatocellular Carcinoma Beyond Milan Criteria: A Prospective, Randomised, Open-Label, Multicentre Phase 2 Trial"

_jcm, 2020, doi:10.3390/jcm9103264_

Round 1
Reviewer 1 Report
The introduction was a bit shallow and short.
Line 66 – Why did the selection criteria specifically include 20-70 years old patients? Why not 18 to 75 years old patients, for example?
Line 79 – You are assuming that the 3-year RFS of the TAC group will be 30% and that of the SRL group will be 60%” Because the sample size is small to begin with, it is possible that an inaccurate patient number may skew the results or conclusion that SRL has better efficacy or outcome.
Line 92 - The statement that “level of MMF was not monitored but the dosage was 0.5–1.5 g/day” is interesting. This is because even if mycophenolate mofetil (MMF) is well-tolerated in the two randomized groups (SRL Vs. TAC), its precise dosage may be important in light of recent studies that report that it has an unacceptable rate of acute allograft rejection in LT recipients receiving MMF alone. However, this report was based on long-term use and not one month use as indicated in this study.
Line 133 – Table 1. Under “CPT Score”, the letters A is misaligned from B and C.
Line 227-228 – The statement that “SRL clearly showed a benefit in prolonging the OS. The main mechanism of prolonging survival seems to be prolonging the tumour-bearing survival after recurrence”. This sentence is ambiguous. Please restate it in a clearer manner
Author Response
Dear reviewer 1,
We thank you for your insightful comments and suggestions that you have provided. We have made every effort to address the concerns that were raised, and we feel the paper is stronger for the inclusion and consideration of these points. Please see the attachment.

Reviewer 2 Report
The authors provide a prospective, randomised, multicentre phase 2 trial compared the oncological outcomes of SRL versus TAC. The study design is very interesting because in comparison to the everolimus trials in liver graft recipients the mTOR inhibitor (here SIR) was combined with MMF but not TAC. This is of special importance because until now there is not CNI-free immunosuppressive regimen approved be FDA or EMA after LT.
It is important, that no savety issues asscociated with SVR had been detected. Nevertheless, the limitied number of patients included is limitating the conclusions. With respect to the phase II trial desgin, however, the number of patients is reasonable.
However, to my understanding, the primary endpoint was not reached, as there was no difference in 3-year RFS and the secondary outcome was the 3-
109 year OS. Therefore, the results should be interpreted with more caution.
The title is not correct
SIR was only assoicated with improved IS after a mulitvariate analysis not defined as primary study end point and most likely underpowered
Only patients with HCC UNOS -T3/4 were included, therefore the title is misleading.
Author Response
Dear reviewer 2,
We thank you for your insightful comments and suggestions that you have provided. We have made every effort to address the concerns that were raised, and we feel the paper is stronger for the inclusion and consideration of these points. Please see the attachment.
